# Analysis of 4-Hydroxyphenyllactic Acid and Other Diagnostically Important Metabolites of α-Amino Acids in Human Blood Serum Using a Validated and Sensitive Ultra-High-Pressure Liquid Chromatography-Tandem Mass Spectrometry Method

**DOI:** 10.3390/metabo13111128

**Published:** 2023-11-03

**Authors:** Pavel D. Sobolev, Natalia A. Burnakova, Natalia V. Beloborodova, Alexander I. Revelsky, Alisa K. Pautova

**Affiliations:** 1Exacte Labs Bioanalytical Laboratory, 20-2 Nauchny Proezd, 117246 Moscow, Russia; sobolevpd@mail.ru (P.D.S.); nburnakova@me.com (N.A.B.); 2Federal Research and Clinical Center of Intensive Care Medicine and Rehabilitology, 25-2 Petrovka Str., 107031 Moscow, Russia; nvbeloborodova@yandex.ru; 3Chemistry Department, Lomonosov Moscow State University, GSP-1, 1-3 Leninskie Gory, 119991 Moscow, Russia; sorbent@yandex.ru

**Keywords:** microbial metabolites, 4-hydroxyphenylacetic acid, 4-hydroxybenzoic acid, phenyllactic acid, phenylpropionic acid, indole-3-acetic acid, indole-3-propionic acid, indole-3-lactic acid, 5-hydroxyindole-3-acetic acid, indole-3-carboxylic acid

## Abstract

The profile of and dynamic concentration changes in tyrosine, phenylalanine, and tryptophan metabolites in blood are of great interest since they could be considered potential biomarkers of different disorders. Some aromatic metabolites, such as 4-hydroxyphenyllactic, 4-hydroxyphenylacetic, phenyllactic, and 4-hydroxybenzoic acids have previously demonstrated their diagnostic significance in critically ill patients and patients with post-COVID-19 syndrome. In this study, a sensitive method, including serum protein precipitation with methanol and ultra-high-pressure liquid chromatography-tandem mass spectrometry (UPLC-MS/MS) detection, was developed and validated for six phenyl- and five indole-containing acids in human serum. The liquid–liquid extraction was also examined, but it demonstrated unsatisfactory results based on analyte recoveries and the matrix effect. However, the recoveries for all analytes reached 100% and matrix effects were not observed using protein precipitation. This made it possible to use deionized water as a blank matrix. The lower limits of quantitation (LLOQs) were from 0.02 to 0.25 μmol/L. The validated method was used for the analysis of serum samples of healthy volunteers (*n* = 48) to reveal the reference values of the target analytes. The concentrations of the most clinically significant metabolite 4-hydroxyphenyllactic acid, which were revealed using UPLC-MS/MS and a previously developed gas chromatography-mass spectrometry method, were completely comparable. The proposed UPLC-MS/MS protocol can be used in the routine clinical practice of medical centers.

## 1. Introduction

The discovery of new biomarkers for various disorders is an important and highly demanded issue in the medicine and related scientific fields [1,2]. Alterations to the metabolite profile may indicate the presence of a pathological process and can be considered subsequently as a diagnostic tool [3]. We have previously demonstrated that a number of metabolites of aromatic α-amino acids are ubiquitous, and their low serum concentrations reflect the integration of the human metabolism with its microbiota [4]. Some of the metabolites can be used as prognostic markers of infectious complications and outcomes in severe patients with various pathologies [5] and in patients with post-COVID-19 syndrome [6]. These metabolites include phenyl-containing microbial and endogenous derivatives of phenylalanine (phenyllactic (PhLA) and phenylpropionic (PhPA) acids) and tyrosine (4-hydroxyphenyllactic (*p*-HPhLA), 4-hydroxyphenylacetic (*p*-HPhAA), 4-hydroxybenzoic (*p*-HBA), and 4-hydroxyphenylpropionic (*p*-HPhPA) acids). These compounds, both together and separately (in particular, *p*-HPhLA), were used to build predictive models in blood serum and cerebrospinal fluid using liquid–liquid extraction or microextraction via packed sorbent coupled with gas chromatography-mass spectrometry (GC-MS) [4,5,6]. The study that has described the results of a univariate model using *p*-HPhLA and two multivariate classification models is of particular interest, because it has demonstrated that aromatic metabolites (one or a number of them) can be used in clinical practice for the prognosis of the outcomes of critically ill patients on the day of admission to the intensive care unit [5]. Thus, the development of a reliable method for the determination of aromatic metabolites is the next step towards their active clinical use. Also, the determination of the concentrations of these metabolites in the blood of healthy people is an important step for both clinical and scientific research.

Tryptophan metabolites have recently attracted increased scientific interest due to their potential role in the participation of the central nervous system processes and the development of various infectious and non-infectious diseases, including socially significant neurodegenerative disorders [7,8,9]. The same sample preparation methods used for phenolic acids (modeling their subsequent simultaneous detection) were applied to detect some microbial and endogenous tryptophan metabolites, particularly indole-containing 5-hydroxyindole-3-acetic (5HIAA), indole-3-lactic (3ILA), indole-3-propionic (3IPA), indole-3-acetic (3IAA), and indole-3-carboxylic (3ICA) acids, in serum and cerebrospinal fluid using GC-MS. The results for liquid–liquid extraction were unsatisfactory because of the irreproducible silylation during the derivatization step, while microextraction using packed sorbent yielded more consistent results; however, only 3IAA was detected in all serum and cerebrospinal fluid samples [10]. We suppose that the reason was the insufficient sensitivity of the developed conditions. Thus, another more sensitive method was required, and ultra-high-performance liquid chromatography coupled to tandem mass spectrometry (UPLC-MS/MS) was chosen.

Various methods were developed for the determination of phenolic and indolic acids in serum or plasma with further HPLC-MS/MS or HPLC-UV analysis [11,12,13]. Sample preparation procedures are required in the same way as for GC-MS since these matrices contain a variety of high-molecular compounds, such as proteins, which might influence then selectivity and specificity of the method. Liquid–liquid extraction [14,15], solid-phase extraction [16,17,18], and protein precipitation [19,20,21,22] are utilized for the sample clean-up before the injection into the chromatographic system. However, it is also important to demonstrate the reliability and suitability of the developed method for obtaining consistent and satisfactory results, so method validation is essential. Both the U.S. Food and Drug Administration (FDA) [23] and European Medicines Agency (EMA) [24] have published guidelines that describe common principles and approaches that are recommended for method validation. Even though there are many studies reporting the determination of phenolic and indolic acids in plasma and serum, only a few of the methods used have been properly validated (Appendix A) [14,20,22,25].

The endogenous content of phenolic and indolic acids in serum or plasma should also be taken into account [26]. To determine the content of an unknown compound in samples, it is necessary to build a calibration curve using a matrix that is the same or similar to that in the samples. However, it is quite a challenging task in the case of endogenous compounds, since the limit of detection (LOD) and the limit of quantitation (LOQ) of the method cannot be less than the lowest concentration of the compound found in healthy donor samples. The content of phenolic and indolic acids in plasma or serum may vary greatly depending on their individual and health conditions [4]; that is why it is important to create a method that would allow one to quantify these compounds at various levels of concentrations, lower or higher than in healthy donor samples. For that purpose, a compound-free matrix might be needed to build a calibration curve. A method for matrix clean-up with charcoal was suggested [20]. It allows for the extraction of indolic acids from plasma before using an obtained surrogate matrix for experiments. Finally, in some cases, it is possible to use simple solvents as an alternative [13,22], which we examined in the present study. The main aim of our study was to develop and validate the conditions for the simultaneous determination of six phenyl- and five indole-containing acids in serum using UPLC-MS/MS and to test the developed method on serum samples of healthy volunteers to reveal reference concentrations of the analytes.

## 2. Materials and Methods

### 2.1. Chemicals and Reagents

Indole-3-carboxylic acid (3ICA, ≥99%), indole-3-acetic acid (3IAA, ≥98%), indole-3-propionic acid (3IPA, ≥99%), indole-3-lactic acid (3ILA, ≥99%), 5-hydroxyindole-3-acetic acid (5HIAA, ≥98%), 3-phenylpropionic acid (PhPA, ≥99%), 3-phenyllactic acid (PhLA, ≥98%), 4-hydroxybenzoic acid (*p*-HBA, ≥99%), 2-(4-hydroxyphenyl)acetic acid (*p*-HPhAA, ≥98%), 4-(3-hydroxyphenyl)propionic acid (*p*-HPhPA, ≥98%), 4-(3-hydroxyphenyl)lactic acid (*p*-HPhLA, ≥97%), indole-3-acetic acid-d_4_ (internal standard, 3IAA-d_4_, ≥98%), and formic acid (≥98%) were purchased from Sigma-Aldrich (Darmstadt, Germany); dimethyl sulfoxide (≥99.9%) was used to dissolve solid analytes from Biochem Chemopharma (Loire, France); methanol, LC-MS (≥99.9%), was used for sample preparation from Sharlau (Barcelona, Spain); acetic acid glacial (≥99.8%) was used for preparation of the mobile phase from Fisher Chemical (Leicestershire, United Kingdom); acetonitrile, HPLC-S gradient grade (≥99.95%), was used for sample preparation and acetonitrile, LC-MS grade (≥99.95%) was used for preparation of the mobile phase from Biosolve (Dieuze, France); ethyl acetate (≥99.8%) and methyl-tert-butyl ether (≥99.9%) were used for liquid–liquid extraction from PanReac AppliChem (Barcelona, Spain); diethyl ether and potassium hydroxide were analytical grade and obtained from Khimmed (Moscow, Russia). HPLC-grade water was prepared using a Milli-Q system (Millipore, Burlington, MA, USA).

### 2.2. Serum Sample Collection

The blood samples from healthy volunteers (*n* = 54) were collected from a peripheral vein into anticoagulant-free test tubes. They were 19 women and 35 men aged from 20 to 67 years. At the time of the examination, volunteers were excluded based on chronic liver and kidney diseases, and there were no general clinical signs of acute inflammation. Serum samples were obtained via blood centrifugation at 1500× *g* for 10 min on the same day. Serum aliquots were poured into disposable Eppendorf tubes, frozen, and stored at −80 °C in the Federal Research and Clinical Center of Intensive Care Medicine and Rehabilitology (Moscow, Russia). The approval of the Local Ethics Committee was obtained (N H01/18, 12 July 2018). UPLC-MS/MS analysis was carried out in the Exacte Labs Bioanalytical Laboratory (Moscow, Russia).

### 2.3. Preparation of Stock and Working Solutions

Stock solutions of analytes and the 3IAA-d_4_ internal standard (IS), with a concentration of 1.0 mg/mL each, were prepared by dissolving the appropriate amount of pure solid standards in dimethyl sulfoxide. Stock solutions were stored at −80 °C until use. Working solutions for calibration standards and quality control (QC) samples were made through the successive dilution of stock solutions with water to obtain concentrations in the range of 2.5–250 μmol/L for *p*-HPhLA, *p*-HPhAA, and PhPA, 0.5–50 μmol/L for *p*-HBA, *p*-HPhPA, and PhLA, 0.2–20 μmol/L for 5HIAA and 3ICA, and 2.0–200 μmol/L for 3ILA, 3IAA, and 3IPA. The working solution of IS was prepared the same way, with a final concentration of 5 μmol/L. To obtain calibration standards and QC samples, working solutions were spiked into 100 μL of water or serum. The final concentration ranges for analytes in samples were 0.25–25 μmol/L for *p*-HPhLA, *p*-HPhAA, and PhPA, 0.05–5 μmol/L for *p*-HBA, *p*-HPhPA, and PhLA, 0.02–2 μmol/L for 5HIAA and 3ICA, and 0.2–20 μmol/L for 3ILA, 3IAA, and 3IPA.

### 2.4. Sample Preparation Conditions

#### 2.4.1. Liquid–Liquid Extraction

An aliquot (10 μL) of the analyte working solution and an aliquot (10 μL) of the IS working solution (5 μmol/L), followed by 50 μL of formic acid solution (10%, *v*/*v*), were added to an aliquot (100 μL) of serum or deionized water. After vortexing the sample, 500 μL of ethyl acetate was added for extraction. The sample was mixed well with the extractant and centrifuged at 2750× *g* for 15 min at 4 °C. Next, an aliquot (400 μL) of organic extract was transferred to the test tube. For the second extraction cycle, 500 μL of ethyl acetate was added to the sample again. After mixing and centrifugation, 400 μL of organic extract was transferred to the same test tube with the first aliquot of extract. Combined extracts were evaporated under the stream of nitrogen (40 L/min) at 40 °C. The dry residue was dissolved in 400 μL of the mobile phase solution (0.2% acetic acid and 5% acetonitrile in water, *v*/*v*), mixed well and transferred to a test plate for further UPLC-MS/MS analysis.

#### 2.4.2. Protein Precipitation

An aliquot (10 μL) of the analyte working solution and an aliquot (10 μL) of the IS working solution (5 μmol/L) were added to an aliquot (100 μL) of serum or deionized water. After vortexing the sample, 400 μL of cooled (4 °C) methanol was added for precipitation, and the sample was mixed well and centrifuged (2750× *g*, 15 min, 4 °C). An aliquot (200 μL) of the supernatant was transferred to a test plate for further UPLC-MS/MS analysis.

### 2.5. UPLC-MS/MS Conditions

The Waters Acquity UPLC I-Class System, consisting of a binary solvent manager (BSM), a sample manager with a flow-through needle (SM-FTN), a column heater-active (CH-A), and a sample organizer (Waters, Milford, MA, USA), was coupled with an AB Sciex QTRAP 5500 mass spectrometer equipped with a Turbo-V™ ion source (AB Sciex, Framingham, MA, USA). All data were acquired and processed using Analyst software v. 1.6.3 (AB Sciex, Framingham, MA, USA). Calculations were performed using Microsoft Excel 2013 (Microsoft, Redmond, WA, USA).

Chromatographic separation was performed on a reverse phase column YMC-Triart C18 (50 mm × 2.0 mm, 1.9 µm). A step-gradient elution program was applied consisting of mobile phases A (0.2% acetic acid in water) and B (0.2% acetic acid in acetonitrile) as follows: 5% B from 0.00 to 4.00 min; 5–35% B from 4.00 to 8.50 min; 35–100% B from 8.50 to 8.55 min; 100% B from 8.55 to 9.50 min; 100–5% B from 9.50 to 9.55 min; 5% B from 9.55 to 10.00 min. The total analysis time was 10 min. The column temperature was 40 °C; and a flow rate of 0.4 mL/min was used. The injection volume was 2 μL, and the autosampler injection needle was washed with acetonitrile/water (1:1, *v*/*v*) after injection. Samples were maintained at +4 °C in the autosampler.

The mass spectrometer equipped with an electrospray ionization (ESI) source was operated in positive and negative ion modes. Nitrogen was used as a nebulizer, as well as a curtain gas. The ion source conditions were set as follows: temperature, 600 °C; ion spray voltage, −4000 V for the negative mode and 4000 V for the positive mode; the nebulizer and heat gas pressure, 90 psi; curtain gas pressure, 40 psi; the collision gas was set to medium flow.

The Scheduled MRM™ Algorithm was used. The multiple-reaction monitoring (MRM) detection window for each analyte was 60 s, and the target scan time was 0.2 s. Optimized analyte-related mass spectrometer parameters are summarized in Table 1.

### 2.6. Method Validation

The conditions of the sample preparation and UPLC-MS/MS analysis were developed and validated in accordance with FDA Guidance for Industry “Bioanalytical Method Validation”, May 2018 [23]. Method validation included the preparation of calibration standards in the appropriate quantitation range to reveal a calibration curve; the preparation of QC samples to assess the precision and accuracy of the assay, including the sensitivity evaluated with the lower limit of quantification (LLOQ); an assessment of the selectivity, carry-over, recovery and matrix effect, chemical stability of the analytes in a matrix, including the autosampler, and short-term, freeze-thaw, long-term stability; and the stability of the stock solution and short-term and long-term stability of working solutions. All validation results are thoroughly described in the Appendix A.

#### 2.6.1. Sensitivity and Selectivity

Selectivity was shown using eight serum samples: six samples from different donors, one hyperlipidemic (2% intralipid solution in pooled serum), and one hemolyzed (5% whole blood in pooled serum) sample. Each sample was analyzed thrice: spiked with both analyte solutions at an LLOQ concentration level and IS solution; spiked with only IS solution; and blank serum sample without an added concentration of analytes and the IS. Sensitivity was evaluated based on how precise and accurate the analyte determination at the LLOQ level was in samples prepared in deionized water (the coefficient of variation (CV) or the relative standard deviation for six samples was no more than 20% and determined concentrations for at least five samples from six were in the range from 80 to 120% of the nominal value). Also, the signal-to-noise ratio was assessed, which was supposed to be no less than 10:1 for the LLOQ and 3:1 the LOD.

#### 2.6.2. Linearity and Carry-Over

Calibration curves were prepared in deionized water instead of serum and consisted of a blank sample (no target analytes or IS), sample spiked with an IS solution, and eight samples spiked with both IS and analytes at different concentration levels. The dependency of the area ratio (absolute detector response to the analytes divided by absolute response to the IS) on the concentration was used in the weighted linear regression analysis with the weighted coefficient 1/x^2^. The calibration curve plot was built using integrated software (Analyst 1.6.3), which also allowed for the calibration curve equations for every analyte and regression coefficient to be exported. The carry-over evaluation consisted of injecting three blank samples after the successive injection of six samples with the highest concentration (the upper limit of quantitation, ULOQ). In the blank samples, the analyte and IS responses should not exceed 20% of the LLOQ and 5% of the mean IS response, respectively.

#### 2.6.3. Accuracy and Precision

The intra- and inter-day accuracy and precision for each analyte were determined at four different concentrations: LLOQ, low QC (LQC), mid QC (MQC) and high QC (HQC); their exact concentrations are described in Appendix A. The intra-day accuracy and precision were determined by six-fold measuring of the respective QC samples within one day. Inter-day accuracy and precision were assessed within three consecutive days. All QCs were prepared based on deionized water, as well as calibration standards.

#### 2.6.4. Recovery and Matrix Effect

The recovery characterizes the efficiency of the extraction procedure and is evaluated based on a comparison of the analyte and IS signal in samples at three concentration levels, spiked before (pre-spike) or after (post-spike) the sample preparation process. The recovery was studied using at least six samples obtained from different donors. Additionally, according to the FDA, it is preferable to include hyperlipidemic and hemolyzed samples in this experiment. The analyte recovery is expressed as a percentage, and its absolute value is not regulated, so it does not need to be 100%. However, it is important that the recoveries of both the analyte and IS are consistent and reproducible (the CV for the pre-/post-spikes of the same concentration is less than 15%). The normalized recovery was calculated using the following formula:(1)Normalized Recovery, %=(Analyte Area in pre-spike − Analyte Area in blank)(IS Area in pre-spike) (Analyte Area in post-spike − Analyte Area in blank)(IS Area in post-spike)∗100%
where pre-spike is the sample that was subjected to the sample preparation procedure; post-spike is the sample that was prepared without adding the analytes and the IS before adding the extractant. The working solutions of analytes and IS were spiked at the final stage of the sample preparation procedure; the blank is the sample without the addition of the analyte working solution.

The matrix effect was assessed using a matrix factor, which was determined by the comparison of the analyte and IS signal in post-spiked samples with samples prepared in the solvent, e.g., water, at three concentration levels. Its normalized value (matrix factor of an analyte divided by the matrix factor of the IS) is not regulated, but CV should be no more than 15%. The normalized matrix factor was calculated using the following formula:(2)Normalized Matrix Factor=(Analyte Area in post-spike − Analyte Area in blank)(IS Area  in post-spike) (Analyte Area in solvent)(IS Area in solvent)
where post-spike is the sample that was prepared without adding the analytes and IS before adding the extractant. Working solutions of analytes and IS were spiked at the final stage of the sample preparation procedure; the blank is the sample without the addition of analyte working solutions; the solvent is the pure solution of the analytes and IS.

#### 2.6.5. Stability

The stability of analytes was investigated at LQC, MQC, and HQC concentrations in three replicates. The stability of analytes in serum was assessed by comparing the mean concentration of analytes in test conditions with the corresponding mean concentration in fresh prepared samples. The stability in the working solution of the analytes and IS was calculated as the ratio of the concentration in test samples to the nominal concentration.

## 3. Results

The study included three stages, namely the selection and development of the sample preparation conditions (Section 3.1), the validation of the selected conditions (Section 3.2), and the analysis of real serum samples of the healthy volunteers to reveal reference values of the target analytes (Section 3.3).

### 3.1. Selection and Development of the Sample Preparation Conditions

As noted in the Introduction Section, the results for liquid–liquid extraction of the tryptophan metabolites using the same conditions as for the phenolic acids were unsatisfactory and irreproducible because two types of silyl derivatives (completely substituted derivatives and derivatives with an unsubstituted hydrogen ion in the indole ring) were obtained during the derivatization step, which was required for the following GC-MS analysis [10]. Thus, we supposed that the use of liquid–liquid extraction coupled with UPLC-MS/MS without derivatization would lead to the reproducible results.

The liquid–liquid extraction included the acidification, extraction, evaporation of the organic extractant, and dilution of dry residue for subsequent UPLC-MS/MS analysis. The variables that were examined during the liquid–liquid extraction development were the following:pH and type of acid (Appendix A);The type of the organic extractant (Appendix A);Evaporation conditions (Appendix A);The type of the solvent, which was used to redissolve the dry residue (Appendix A);The number of extraction cycles (Appendix A).

The parameters that were used to evaluate the efficacy of the selected conditions were the recovery and matrix factor, with the desired recovery reaching 100% at the same time as matrix factor reaching 1.00.

Neutral and acidic pH values, created by the addition of formic or sulfuric acids to the serum solution, were examined (Appendix A). Acids were needed to decrease the pH of serum samples to 2 when most of the studied analytes become uncharged due to the fact that their *p*K_a_ values were from 3.5 to 4.8. Thus, the extraction of the analytes is increased. Results for the recovery and matrix factor in neutral pH were absolutely unsatisfactory, while acidic conditions demonstrated comparable recoveries for most analytes with an unsatisfactory matrix factor for 3ILA and 3IAA. However, formic acid revealed slightly better results for the matrix factor for 3ILA and 3IAA compared to sulfuric acid; thus, it was chosen for the further experiments.

Diethyl ether, methyl-*tert*-butyl ether, and ethyl acetate were used as organic extractants (Appendix A). The highest recoveries were achieved for ethyl acetate, with the mean values from 43 to 75%. The matrix factors for the most analytes were comparable and were within 0.85–1.09 for all extractants; the matrix factor for 3IPA was less satisfactory and had a mean value of 0.70 for all extractants; for 3ILA and 3IAA, the values of the matrix factor were still unsatisfactory and had mean values of 0.34 and 0.38, respectively.

The influence of the evaporation conditions was studied. The following sample preparation methods were investigated: evaporation of the extract by heating to dryness, evaporation of the extract in the presence of 50 μL of deionized water (evaporation without the formation of a dry residue), and evaporation to dryness without heating (Appendix A). Changing the evaporation conditions led to an increase in the recoveries of some compounds (*p*-HPhLA, *p*-HBA, PhLA, and 3ILA), but did not improve the matrix effect of 3ILA and 3IAA.

The mobile phase, acetonitrile or methanol, was examined to redissolve the dry residue after complete evaporation (Appendix A). The recoveries using methanol were better for most analytes than the other conditions, while the matrix factor remained unsatisfactory for 3ILA and 3IAA.

Single, double, and triple extractions were studied experimentally (Appendix A). Significant differences were observed between single and double extractions, while no difference was detected between double and triple extraction. The recoveries reached 80–90% for most analytes, except 5HIAA, while the matrix factor remained unsatisfactory for 3ILA and 3IAA.

At this stage, we decided to stop searching for the further liquid–liquid extraction conditions and made the conclusion that liquid–liquid extraction was not suitable for the simultaneous determination of phenolic and indolic compounds. We also took our previous results into account when developing the GC-MS conditions [10] and now suppose that our results were unsatisfactory not only because of the two types of silyl derivatives during derivatization, but also because of the incomplete extraction.

The protein precipitation is undoubtedly one of the fastest sample preparation methods, and we examined the two most common organic precipitators, namely methanol [13] and acetonitrile [20]. Although acetonitrile was more preferable as it was a part of the mobile phase, the best results were obtained with methanol (Appendix A), and it was chosen in final sample preparation conditions, which are listed above in Section 2.4.2. The recoveries were 91–101% with a CV less than 8%, and the matrix factor was within 0.90–1.09 with a CV less than 9%.

Important information that may be of interest for the further use of this method in real practice is the duration of sample preparation and other preparatory steps. The samples were thawed at room temperature for about 2 h and vortexed before sample preparation. As for the analysis time, it takes about 1 h to prepare a 96-well plate, taking into account all stages of the sample preparation: aliquoting takes from 5 to 25 min (depending on the number of samples), adding the working solution of IS and methanol takes about 10 min, and centrifugation and transfer of the supernatant into a 96-well plate takes about 20 min.

### 3.2. Validation of the UPLC-MS/MS Method with Protein Precipitation

A QTRAP mass spectrometer in the MRM mode was used under positive and negative ESI conditions. The precursor ions for indole-containing analytes were [M + H]^+^, and for phenyl-containing analytes, they were [M − H]^−^. For indole-containing analytes, the formation of the precursor ions [M − H]^−^ was also observed, and MRM transitions were found. Unfortunately, for the studied indolic acids, the response in the negative mode was lower than in the positive mode. The precursor ions for phenyl-containing analytes were not observed in the mass spectrum in the positive scanning mode. Two product ions were chosen based on the response for each analyte: one was used in quantitative calculations (the first product ion for every compound in Table 1) and the other for additional qualitative identification of the compound (the second product ion for every compound in Table 1). PhPA has only one selective MRM transition. For this reason, there is no additional transition in the UPLC-MS/MS method. Mass spectra of the Q1-scan mode based on the first quadrupole and product ion scan modes of all analytes and the IS are presented in Appendix A. The ion source parameters, such as temperature, ion spray voltage, and gas flows, were optimized using the flow injection analysis (FIA) to achieve the best signal-to-noise ratio for all analytes. The final mass spectrometric conditions are given in Section 2.5.

Since the tyrosine, phenylalanine, and tryptophan metabolites are endogenous small polar molecules, we used the reversed phase chromatography and optimized the composition of the mobile phase to obtain better separation of the compounds. A chromatography column, YMC-Triart C18, was used in the study. Gradient elution with 0.2% acetic acid in water (mobile phase A) and 0.2% acetic acid in acetonitrile (mobile phase B) allowed the symmetrical peak shapes and reproducible retention to be to achieved. For the first four minutes of the chromatographic method, an isocratic flow of the mobile phase was set: the ratio of mobile phases A and B was 95:5. During this time, the four most polar compounds (*p*-HPhLA, *p*-HBA, *p*-HPhAA, and 5HIAA) were eluted. Between 4.00 and 8.50 min the concentration of the mobile phase B was increased from 5 to 35%, and eight remaining compounds were washed out in the following order: *p*-HPhPA, PhLA, 3ILA, 3ICA, 3IAA, 3IAA-d_4_, PhPA, and 3IPA. The last compound (3IPA) was eluted at 8.30 min; therefore, after 8.50 min, the concentration of the mobile phase B was raised drastically from 35 to 100% and maintained at this level for the next 0.95 min. This step was needed for cleaning and regenerating the column and washing out all unrelated substances that could still be retained in a processed sample. It also helped to avoid carry-over effects (Appendix A). Finally, the ratio of mobile phases A and B changed back to the starting conditions (95:5), and the column was equilibrated for the last 0.45 min of the chromatographic method before the next injection. The analytes and IS were eluted within 9 min, which allowed us to develop the fast method with a total run time of 10 min. The retention times of all compounds ranged from 2.1 to 8.3 min and remained constant during method validation and analysis of healthy volunteers’ samples, as well as the column back pressure. A representative chromatogram of all analytes is shown in Figure 1. Although the chromatographic peaks of *p*-HPhAA and 5HIAA, *p*-HPhPA and PhLA, and PhPA and 3IPA partially or fully overlap, it did not affect any validation parameters because the analytes had different MRM transitions. In particular, two compounds from our list of the analytes, *p*-HPhPA and PhLA, have equal molecular masses. Despite having the same retention time (5.8 min) and *m*/*z* value of the precursor ion (164.9), they have different product ions (93.0 and 59.0 for *p*-HPhPA; 103.0 and 73.0 for PhLA). Therefore, the isobaric nature of these analytes did not prevent their correct and independent MS/MS determination while using a common C18 column.

Selecting a suitable IS that fully compensates for the matrix effects for all analytes is an important and difficult task. The stable isotope-labeled internal standard 3IAA-d_4_ has a similar chemical structure to the analytes, thus compensating for losses of the analytes during the sample preparation step and fully leveling out matrix effects. 3IAA-d_4_ in the negative mode was used as an IS for phenyl-containing acids and in the positive mode—for indole-containing acids.

Due to the fact that the studied phenyl- and indole-containing acids are endogenous compounds, obtaining analyte-free serum is a very difficult task. In this study, we applied a surrogate matrix approach, using deionized water instead of the serum. To be able to do so, it is necessary to demonstrate that the recovery and matrix effect in water are comparable to those in serum [26]. We studied the recovery and matrix effect of the analytes and IS, and it is shown that the values of these two parameters for all compounds were close to 100% (Appendix A) and one (Appendix A), respectively. These facts allowed us to use the deionized water instead of serum for the preparation of calibration standards and QC samples.

Selectivity is a special parameter in the validation, which demonstrates the absence of peak interference from unrelated compounds present in the matrix with analytes. The chromatograms of the blank matrix and the QC samples at the LLOQ concentration prepared with deionized water, as well as the chromatograms of blank serum samples of healthy volunteers, are presented in the Appendix A. There were no overlapping peaks in the chromatograms of blank samples, both water and serum. Chromatograms of serum samples contained only peaks of the studied compounds (*p*-HPhLA, *p*-HPhAA, PhPA, PhLA, 5HIAA, 3ILA, 3IAA, 3IPA), which have an endogenous nature. The analytical parameters of the developed method, such as calibration curve equations and LOD and LLOQ values are presented in Table 2. The calibration ranges of the validated method are presented in Section 2.3. The carry-over of all analytes and the IS was within the acceptable limits (Appendix A).

As a part of the validation, the intra- and inter-day accuracy and precision of the method were studied at four different concentrations. The calculated concentrations of LQC, MQC, and HQC were within the acceptable limits of 15% of the nominal concentration and 20% for the LLOQ. Precision (CV, %) did not exceed 20% for the LLOC sample and 15% for other quality control samples. All of the above-listed facts demonstrated that the developed method was accurate, reliable, and reproducible (Appendix A).

The stability of analytes in serum, including the autosampler stability (+4 °C for 336 h), short-term stability in serum (room temperature for 24 h), three-freeze–thaw-cycle stability in serum, and long-term stability in serum (−80 °C for 175 days), were studied (Appendix A). The short-term stability of working solutions of the analytes at room temperature (24 h) and the long-term stability of working solutions of the analytes and IS (+4 °C for 9 days) are also demonstrated in Appendix A. The stock solution stability at 4 °C was 1142 days (Appendix A).

There are a number of articles describing the determination of compounds of interest in serum or plasma; however, only few of them include a full validation of the method used [14,20,22,25]. The UPLC-MS/MS method developed during this study was properly validated in accordance with FDA Guidance for Industry “Bioanalytical Method Validation”, May 2018 [23]. The sample preparation process requires only protein precipitation with an organic solvent with subsequent injection of the supernatant into the chromatographic system. Thus, this sample pretreatment procedure is quick and simple, opposed to the ones that include liquid–liquid extraction [14] or solid-phase extraction [12]. Moreover, the developed method proved to be sensitive and selective without the necessity of using specific chromatographic columns [13]. The key point was to achieve recoveries of the analytes close to 100% and matrix effects equal to one for all of them, so that using water instead of native or cleaned-up plasma [20] as a surrogate matrix would be possible. However, although the sample volume for this method is only 100 μL, there are studies that show the prospect of using even less matrix for the analysis [20]. While it is not a disadvantage of the developed method for the determination of analytes in human serum or plasma of adult people, it may be crucial in cases of children or neonates when the available volume of the sample might be limited.

### 3.3. Analysis of Serum Samples of the Healthy Volunteers

The validated method was applied for the diagnostically important issue, namely the determination of reference values of analytes in the serum samples of the adult healthy volunteers (*n* = 48). All samples were analyzed in triplicate; the results are accumulated in Table 3 and Appendix A. The following analytes were quantitatively measured in all samples: 5HIAA, 3ILA, 3IAA, PhLA, and *p*-HPhLA; 3IPA, PhPA, and *p*-HPhAA were measured in 92, 79, and 71% of cases; *p*-HBA and *p*-HPhPA were measured in 4% of cases; and 3ICA was not measured in any cases.

To analyze the data obtained, we needed to compare it with some previous studies. As for phenyl-containing acids, there are several studies describing their normal and abnormal levels. In a 2018 study, the median reference concentration of PhPA and *p*-HPhLA obtained using GC-MS in serum samples of 40 healthy volunteers was 0.5 μmol/L [4]. In a recent study, the data also obtained using GC-MS on the same 48 healthy volunteers as in our study (Table 3) demonstrated the presence of PhPA, PhLA, *p*-HPhAA, and *p*-HPhLA in serum samples with different frequencies [6]. Only *p*-HPhLA was measured in all 48 samples. As *p*-HPhLA is the most diagnostically important metabolite from our set of analytes and tends to significantly increase in critically ill patients, we decided to compare its concentrations obtained using two methods and accumulated these data in Appendix A. According to the data obtained, we can state that the GC-MS and UPLC-MS/MS data are completely comparable. Moreover, two methods were validated in similar concentration ranges, and we can assume that the clinical results that described the prognostic significance of *p*-HPhLA in critically ill patients and were previously reported using GC-MS [5] will be applicable for UPLC-MS/MS. Considering the excellent validation characteristics of the developed UPLC-MS/MS method in comparison with GC-MS, we believe that we are as close as possible to introducing our method into the routine clinical practice of medical centers that have the appropriate equipment.

PhLA and *p*-HPhAA were rarely quantified in healthy donors via GC-MS, but also tended to be highly elevated in critically ill patients and demonstrated prognostic significance in some clinical studies [5]. The developed UPLC-MS/MS method allowed us to measure their normal concentrations in all samples for PhLA and in 34 of 48 healthy volunteers in comparison with the GC-MS method with insufficient sensitivity.

PhPA was quantitatively measured in healthy donors using GC-MS and completely disappeared in critically ill patients because of their gut microbiota dysbiosis. However, it was impossible to access its prognostic significance because of the insufficient sensitivity of GC-MS method [4]. PhPA concentrations were measured in 38 from 48 healthy volunteers using the developed UPLC-MS/MS method, and considering its sensitivity, we can assume that the future clinical studies will reveal its diagnostic significance.

*p*-HBA recently demonstrated its diagnostic significance in patients with post-COVID-19 syndrome [6]. The data obtained via GC-MS with the LLOQ level of 0.5 μmol/L demonstrated the absence of this metabolite in healthy donors. Interestingly, we detected this metabolite at higher levels than the LOQ level of 0.05 μmol/L in two samples and higher than the LOD of 0.015 μmol/L, but below the LOQ in four samples using the UPLC-MS/MS method; thus, the appearance of this metabolite in patients may indicate pathological processes, which should be studied in the future.

Undoubtedly excellent data were obtained using the developed UPLC-MS/MS method for the indole-containing acids, namely 5HIAA, 3ILA, 3IAA, and 3IPA, which were measured in all serum samples for 5HIAA, 3ILA, and 3IAA and in 44 of 48 samples for 3IPA. These data will be used in future clinical studies as reference values.

3ICA was not quantitatively measured in healthy volunteers in this study even with the LLOQ level of 0.02 μmol/L. However, 3ICA was detected in 73% of cases at levels higher than the LOD; thus, we can assume that this metabolite, like most of our analytes, should be present in healthy people at low concentrations, reflecting the normal microbiota function. A retrospective analysis by GC-MS demonstrated that this metabolite was found in 17 of 288 (6%) serum samples of critically ill patients [10]. Thus, we suppose that 3ICA may be typical for some pathological processes, similar to *p*-HBA.

However, our subsequent research will attempt to reduce the LOQ levels of analytes in serum that were not measured in this study (*p*-HBA, *p*-HPhPA, and 3ICA) and test new conditions for another important biological matrix, namely cerebrospinal fluid.

## 4. Conclusions

Phenyl- and indole-containing metabolites of tyrosine, phenylalanine, and tryptophan are potential biomarkers of different disorders and their simultaneous determination in human blood serum is a challenging issue because of their different chemical structures and endogenous content in human blood. All analytes were separated on a C18 chromatography column, except two isobaric compounds, *p*-HPhPA and PhLA. However, they had different product ions and were successfully analyzed using MRM-mode. Despite liquid–liquid extraction and protein precipitation being examined as sample preparation methods, only protein precipitation demonstrated acceptable results based on almost 100% recoveries and no matrix effect. Moreover, the results based on recoveries and the matrix effect allowed us to use deionized water instead of analyte-free blood serum for the validation experiments. Thus, in this study we developed and validated a sensitive UPLC-MS/MS method for the simultaneous quantification of phenyl- and indole-containing metabolites. The main goal was to test the developed protocol for the analysis of the healthy volunteers’ serum, and we suppose that this goal was successfully achieved as we managed to quantify most analytes. All data on the validation and analysis of the real samples are available in the Appendix A and can be used by the researchers in their future clinical studies as reference values. We believe that our protocol will be applied in clinics for the analysis of the diagnostically significant *p*-HPhLA and other potentially important aromatic metabolites of microbial and endogenous origin.

## Figures and Tables

**Figure 1 metabolites-13-01128-f001:**
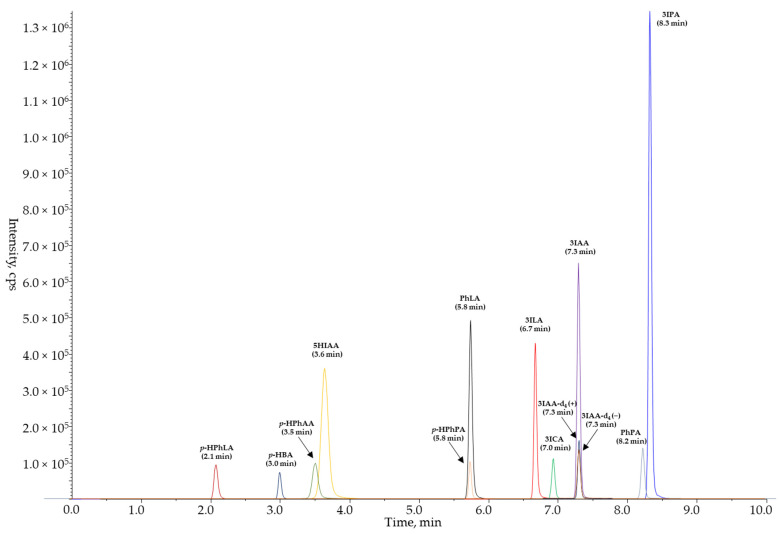
The typical chromatogram of 11 target compounds in the calibration standard with the highest concentration (ULOQ) and the IS. The chromatogram is the sum of mass chromatograms constructed from the MRM transitions of the compounds: 180.9 > 134.0 for *p*-HPhLA (−), 136.9 > 65.1 for *p*-HBA (−), 150.9 > 79.0 for *p*-HPhAA (−), 148.9 > 105.0 for PhPA (−), 164.9 > 93.0 for *p*-HphPA (−), 164.9 > 103.0 for PhLA (−), 192.0 > 146.1 for 5HIAA (+), 206.0 > 118.1 for 3ILA (+), 162.1 > 116.0 for 3ICA (+), 176.1 > 130.1 for 3IAA (+), 190.1 > 77.0 for 3IPA (+), 180.1 > 133.1 for IS 3IAA-d_4_ (+), 178.0 > 134.1 for IS 3IAA-d_4_ (−).

**Table 1 metabolites-13-01128-t001:** Summary of MS/MS parameters optimized for indole- and phenyl-containing analytes and indole-3-acetic acid-d_4_ as the internal standard.

Compound	Structure	Abbreviation	Precursor ion	Product ion	ESI	DP (V) *	CE (V) **	CXP (V) ***
4-Hydroxyphenyllactic Acid	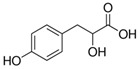	*p*-HPhLA	180.9	134.0	-	−95	−22	−7
119.0	-	−95	−24	−13
4-Hydroxybenzoic Acid	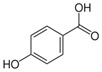	*p*-HBA	136.9	65.1	-	−90	−40	−9
93.0	-	−90	−22	−11
4-Hydroxyphenylacetic Acid	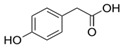	*p*-HPhAA	150.9	79.0	-	−40	−24	−9
107.0	-	−40	−16	−5
Phenylpropionic Acid	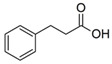	PhPA	148.9	105.0	-	−65	−22	−10
4-Hydroxyphenylpropionic Acid	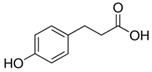	*p*-HPhPA	164.9	93.0	-	−70	−16	−11
59.0	-	−70	−16	−9
Phenyllactic Acid	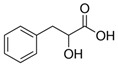	PhLA	164.9	103.0	-	−85	−22	−13
73.0	-	−85	−24	−9
5-Hydroxyindole-3-acetic Acid	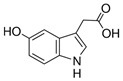	5HIAA	192.0	146.1	+	96	23	10
118.0	+	96	39	14
Indole-3-lactic Acid	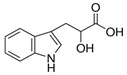	3ILA	206.0	118.1	+	86	50	6
130.1	+	86	39	6
Indole-3-carboxylic Acid	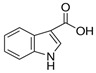	3ICA	162.1	116.0	+	80	29	12
118.1	+	80	19	14
Indole-3-acetic Acid	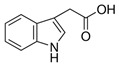	3IAA	176.1	130.1	+	90	52	6
103.0	+	90	43	12
Indole-3-propionic Acid	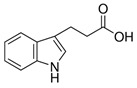	3IPA	190.1	77.0	+	80	80	10
103.1	+	80	47	12
Indole-3-acetic Acid-d_4_	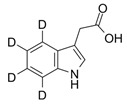	3IAA-d_4_	180.1	133.1	+	91	23	8
178.0	134.1	-	−85	−16	−7

* Declustering potential; ** collision energy; *** collision cell exit potential.

**Table 2 metabolites-13-01128-t002:** Summary of the analytical parameters for the developed sample preparation conditions and UPLC-MS/MS analysis for indole- and phenyl-containing acids.

Compound	Regression Equation	r	LOD (μmol/L)	LLOQ (μmol/L)	ULOQ (μmol/L)	RT (min)
*p*-HPhLA	y = 0.0281 x − 0.00192	0.9993	0.03	0.25	25	2.1
*p*-HBA	y = 0.121 x + 0.00194	1.0000	0.02	0.05	5.0	3.0
*p*-HPhAA	y = 0.0399 x + 0.000930	1.0000	0.08	0.25	25	3.5
PhPA	y = 0.0202 x + 0.00131	0.9994	0.06	0.25	25	8.2
*p*-HPhPA	y = 0.0886 x − 0.000787	0.9998	0.02	0.05	5.0	5.8
PhLA	y = 0.473 x − 0.00294	1.0000	0.004	0.05	5.0	5.8
5HIAA	y = 1.67 x + 0.00565	0.9991	0.003	0.02	2.0	3.6
3ILA	y = 0.0658 x + 0.00249	0.9984	0.03	0.20	20	6.7
3ICA	y = 0.193 x + 0.000938	0.9977	0.006	0.02	2.0	7.0
3IAA	y = 0.113 x + 0.000724	0.9985	0.02	0.20	20	7.3
3IPA	y = 0.200 x + 0.00559	0.9980	0.02	0.20	20	8.3

**Table 3 metabolites-13-01128-t003:** Concentrations of indole- and phenyl-containing acids in the serum samples of the healthy volunteers (*n* = 48), μmol/L.

Compound	*p*-HPhLA	*p*-HBA	*p*-HPhAA	PhPA	*p*-HPhPA	PhLA	5HIAA	3ILA	3ICA	3IAA	3IPA
Median	1.21	<0.015	0.32	0.46	<0.015	0.32	0.078	1.07	<0.02	1.82	1.36
25% Quartile	0.96	<0.015	<0.25	0.27	<0.015	0.25	0.064	0.84	<0.006	1.51	0.77
75% Quartile	1.56	<0.015	0.46	0.72	<0.015	0.39	0.093	1.27	<0.02	2.38	2.09
Minimum	0.76	<0.015	<0.075	<0.06	<0.015	0.19	0.028	0.56	<0.006	0.77	<0.2
Maximum	2.41	0.14	1.43	3.94	0.12	0.66	3.5	4.5	<0.02	4.81	8.08

## Data Availability

Not applicable.

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
