# Peer review of "Analysis of 4-Hydroxyphenyllactic Acid and Other Diagnostically Important Metabolites of α-Amino Acids in Human Blood Serum Using a Validated and Sensitive Ultra-High-Pressure Liquid Chromatography-Tandem Mass Spectrometry Method"

_metabolites, 2023, doi:10.3390/metabo13111128_

Round 1
Reviewer 1 Report
Comments and Suggestions for Authors
Sobolev et al. submitted the manuscript, "Analysis of 4-Hydroxyphenyllactic Acid and Other Diagnostically Important Metabolites of α-Amino Acids in Blood Serum Using Validated and Sensitive HPLC-MS/MS Method" is an interesting study where the author investigated a method development for phenyl indole acids using tandem mass spectrometry (HPLC-MS/MS) methodology.
The following minor points need to be revised:
1. The HPLC method development needs to be revised and written in a more descriptive way, as the authors considered indole and phenyl acids, where Indole and phenyl acids somewhat have distinctive chemical structures, but some of them have isobaric nature (as compounds) and, therefore, can pose challenges in developing a method with a reasonable resolution. Therefore, generally, researchers used columns (specific columns) to study such a diverse set of structural compounds, where few of them are isobaric compounds.
2. The author chose a C18 column and ran a method with a step gradient elution method (where solvent A is (0.2% acetic acid in water) and B (0.2% acetic acid in acetonitrile)
5% B from 0.00 to 4.00 min;
5-35% B from 4.00 to 8.50 min;
35-100% B from 8.50 to 8.55 min;
100% B from 8.55 to 9.50 min;
100-5% B from 9.50 to 9.55 min;
5% B from 9.55 to 10.00 min.
However, the last chromatogram peak was 8.3 mins, from Figure 1. By any chance, was the author trying to clean (wash) or integrate these steps to clean the column at the later end of the method run? Please write it clearly or in a better way.
Author Response
Reviewer 1
Sobolev et al. submitted the manuscript, "Analysis of 4-Hydroxyphenyllactic Acid and Other Diagnostically Important Metabolites of α-Amino Acids in Blood Serum Using Validated and Sensitive HPLC-MS/MS Method" is an interesting study where the author investigated a method development for phenyl indole acids using tandem mass spectrometry (HPLC-MS/MS) methodology.
The following minor points need to be revised:
- The HPLC method development needs to be revised and written in a more descriptive way, as the authors considered indole and phenyl acids, where Indole and phenyl acids somewhat have distinctive chemical structures, but some of them have isobaric nature (as compounds) and, therefore, can pose challenges in developing a method with a reasonable resolution. Therefore, generally, researchers used columns (specific columns) to study such a diverse set of structural compounds, where few of them are isobaric compounds.
The answer: We thank the reviewer for his high assessment of our work, for his time and suggested changes. We have revised the relevant section of the manuscript (Section 3.2) in a more descriptive way and hope this will improve the quality of the material presented. Regarding the question about isobaric compounds, two compounds from our list of the analytes, p-HPhPA and PhLA, have equal molecular masses. Despite having the same retention time (5.8 min) and m/z value of the precursor ion (164.9), they have different product ions (93.0 and 59.0 for p-HPhPA; 103.0 and 73.0 for PhLA). Therefore, the isobaric nature of these analytes did not prevent their correct and independent MS/MS determination, while using a common C18 column. We have also added this information into the Section 3.2 (lines 399-403)
- The author chose a C18 column and ran a method with a step gradient elution method (where solvent A is (0.2% acetic acid in water) and B (0.2% acetic acid in acetonitrile)
5% B from 0.00 to 4.00 min;
5-35% B from 4.00 to 8.50 min;
35-100% B from 8.50 to 8.55 min;
100% B from 8.55 to 9.50 min;
100-5% B from 9.50 to 9.55 min;
5% B from 9.55 to 10.00 min.
However, the last chromatogram peak was 8.3 mins, from Figure 1. By any chance, was the author trying to clean (wash) or integrate these steps to clean the column at the later end of the method run? Please write it clearly or in a better way.
The answer: We thank the reviewer for this comment and we have added more descriptive information describing the gradient mode conditions into the Section 3.2 (lines 371-397). Interrupting the gradient part of the method and switching to washing from 8.5 to 8.55 minutes allowed us to reduce the duration of the method and make it more express. Other ways for integrating final elution of the last peak (3IPA) and washing the column were considered, but it was not possible to get better separation of the compounds.
Reviewer 2 Report
Comments and Suggestions for Authors
The manuscript entitled “Analysis of 4-Hydroxyphenyllactic Acid and Other Diagnostically Important Metabolites of α-Amino Acids in Blood Serum Using Validated and Sensitive HPLC-MS/MS Method” has been reviewed carefully for publication. Authors have established a sensitive MS method to determine the Metabolites of α-Amino Acids in Blood Serum and validated the method. Overall, the manuscript is interesting, the methodology is adequately described, and the manuscript has new information and can be accepted in its present form. I don’t have any specific comments on methodology but I have a few comments as mentioned below:
1. Authors need to modify the title of the manuscript i.e. include human before the blood serum for clear understanding.
2. Can the author provide the chemical structures of each analyte in the revised version?
3. Authors should include the chromatogram of the blank matrix in Fig 1.
4. The formatting and grammatical errors should be verified again throughout the manuscript
Comments on the Quality of English LanguageThe formatting, and grammatical errors should be verified again throughout the manuscript.
Author Response
Reviewer 2
The manuscript entitled “Analysis of 4-Hydroxyphenyllactic Acid and Other Diagnostically Important Metabolites of α-Amino Acids in Blood Serum Using Validated and Sensitive HPLC-MS/MS Method” has been reviewed carefully for publication. Authors have established a sensitive MS method to determine the Metabolites of α-Amino Acids in Blood Serum and validated the method. Overall, the manuscript is interesting, the methodology is adequately described, and the manuscript has new information and can be accepted in its present form. I don’t have any specific comments on methodology but I have a few comments as mentioned below:
- Authors need to modify the title of the manuscript i.e. include human before the blood serum for clear understanding.
The answer: We thank the reviewer for his high assessment of our work, for his time and suggested changes. We have added “Human” into the Title.
- Can the author provide the chemical structures of each analyte in the revised version?
The answer: We have added all chemical structures of the analytes into Table 1.
- Authors should include the chromatogram of the blank matrix in Fig 1.
The answer: The chromatograms of the blank matrix are presented in supplementary (Table S5). The background signal in the blank sample has different values for different analytes (from 10 to 1000 cps). We suppose if the chromatograms are summarized and presented similarly to the chromatogram in Figure 1, it will be difficult to comprehend
- The formatting and grammatical errors should be verified again throughout the manuscript.
The answer: We have tried to correct formatting and grammatical errors in the article. Please, see the revised version of the manuscript
Reviewer 3 Report
Comments and Suggestions for Authors
I have read your paper with great interest and I commend you for your work on developing and validating a HPLC-MS/MS method for the analysis of phenyl- and indole-containing metabolites in human serum.
These metabolites are potential biomarkers of various disorders and their reliable quantification is important for clinical and scientific research.
Your paper is well-written, clear, and informative. You have followed the appropriate guidelines for method validation and presented the results in a comprehensive and transparent manner. You have also applied your method to real serum samples of healthy volunteers and revealed the reference values of the target analytes, which is valuable for future studies.
I have only minor comments and suggestions for improvement:
- Please explain why you used deionized water as a blank matrix instead of serum. How did you ensure that there was no interference from endogenous compounds in serum?
- Please compare your method with other existing methods for the determination of phenolic and indolic acids in serum or plasma. What are the advantages and limitations of your method?
- Please provide some examples or references of how your method can be used in clinical practice or research. What are the potential applications and implications of your method?
Author Response
Reviewer 3
I have read your paper with great interest and I commend you for your work on developing and validating a HPLC-MS/MS method for the analysis of phenyl- and indole-containing metabolites in human serum.
These metabolites are potential biomarkers of various disorders and their reliable quantification is important for clinical and scientific research.
Your paper is well-written, clear, and informative. You have followed the appropriate guidelines for method validation and presented the results in a comprehensive and transparent manner. You have also applied your method to real serum samples of healthy volunteers and revealed the reference values of the target analytes, which is valuable for future studies.
I have only minor comments and suggestions for improvement:
- Please explain why you used deionized water as a blank matrix instead of serum.
The answer: We thank the reviewer for his high assessment of our work, for his time and suggested changes. We have mentioned this information in the Introduction Section (lines 91-104): “To determine the content of an unknown compound in samples, it is necessary to build a calibration curve using a matrix that is the same or similar to that in the samples. However, it is quite a challenging task in case of endogenous compounds since the limit of detection (LOD) and the limit of quantitation (LOQ) of the method cannot be less than the lowest concentration of the compound found in healthy donors' samples. Content of phenolic and indolic acids in plasma or serum may vary greatly depending on their individual and health conditions [4], that is why it is important to create a method that would allow one to quantify these compounds at various levels of concentration, lower or higher than in healthy donor's samples. For that purposes, compound-free matrix might be needed to build a calibration curve. A method for matrix clean-up with charcoal was suggested [22] . It allows the extraction of indolic acids from plasma before using an obtained surrogate matrix for experiments. Finally, in some cases it is possible to use simple solvents as an alternative [15, 24] , which we examined in the present study.“ Also this information was mentioned in Section 3.2 (lines 419-429): “Due to the fact that the studied phenyl- and indole-containing acids are endogenous compounds, obtaining analyte-free serum is a very difficult task. In this study we applied a surrogate matrix approach, using deionized water instead of the serum. To be able to do so it is necessary to demonstrate that the recovery and matrix effect in water are comparable to the ones in serum [28]. We studied the recovery and matrix effect of the analytes and IS, and it is shown that the values of these two parameters for all compounds were close to 100% (Table S3) and one (Table S4), respectively. These facts allowed us to use the deionized water instead of serum for the preparation of calibration standards and QC samples.”
- How did you ensure that there was no interference from endogenous compounds in serum?
The answer: Selectivity is a special parameter in the validation which demonstrates the absence of the interferences of the analytes. According to FDA guidelines “During method development, the sponsor should verify that the substance being measured is the intended analyte to minimize or avoid interference. Selectivity of the method is routinely demonstrated by analyzing blank samples of the appropriate biological matrix (e.g., plasma) from multiple sources. Depending on the intended use of the assay, the impact of hemolyzed samples, lipemic samples, or samples from special populations can be included in the selectivity assessment. When using liquid chromatography/mass spectrometry (LC/MS) methods, the sponsor or applicant should determine the effects of the matrix on ion suppression, ion enhancement, or extraction efficiency. Internal standards should be assessed to avoid interference with the analyte. Potential interfering substances in a biological matrix include endogenous matrix components such as metabolites, decomposition products – and from the actual study – concomitant medications and other xenobiotics.” We also have assessed the selectivity, which is described in Section 2.6.1. Sensitivity and Selectivity. The results of the selectivity are mentioned and expanded in Results and Discussion (lines 430-439): Selectivity is a special parameter in the validation which demonstrates the absence of peak interference from unrelated compounds present in the matrix with analytes. The chromatograms of the blank matrix and the QC samples at LLOQ concentration prepared with deionized water, as well as chromatograms of blank serum samples of healthy volunteers, are presented in the supplementary (Table S5). There were no overlapping peaks in the chromatograms of blank samples , both water and serum. Chromatograms of serum samples contain only peaks of the studied compounds (p-HPhLA, p-HPhAA, PhPA, PhLA, 5HIAA, 3ILA, 3IAA, 3IPA) which have endogenous nature.”
- Please compare your method with other existing methods for the determination of phenolic and indolic acids in serum or plasma.
The answer: We compared our method to others that were described in the Introduction section and added a paragraph dedicated to drawing differences between them, at the end of Section 3.2. Validation of the UPLC-MS/MS method with protein precipitation (lines 459-475): "There are a number of articles describing determination of compounds of interest in serum or plasma, however, only few of them include full validation of the used method [16, 22, 24, 27]. The UPLC-MS/MS method developed during this study was properly validated in accordance with FDA Guidance for Industry “Bioanalytical Method Validation”, May 2018 [25]. The sample preparation process requires only protein precipitation with organic solvent with subsequent injection of the supernatant into the chromatographic system. Thus, this sample pretreatment procedure is quick and simple, opposed to the ones that include liquid-liquid extraction [16] or solid-phase extraction [14]. Moreover, the developed method proved to be sensitive and selective without necessity to use specific chromatographic columns [15]. The key point was to achieve recoveries of the analytes close to 100% and matrix effects equal to one for all of them, so that using water instead of native or cleaned-up plasma [22] as a surrogate matrix would be possible. However, although the sample volume for this method is only 100 μL, there are studies that show the prospect of using even less matrix for analysis [22]. While it is not a disadvantage of the developed method for determination of analytes in human serum or plasma of adult people, it may be crucial in cases of children or neonates when the available volume of the sample might be limited."
- What are the advantages and limitations of your method? строчки 480-482.
The answer: The advantages of our method have been revised and accumulated in a new version of the Conclusion Section “Phenyl- and indole-containing metabolites of tyrosine, phenylalanine and tryptophan are potential biomarkers of different disorders and their simultaneous determination in human blood serum is a challenging issue because of their different chemical structures and endogenous content in human blood. All analytes were separated on a C18 chromatography column except two isobaric compounds p-HPhPA and PhLA. However, they had different product ions and were successfully analyzed using MRM-mode. Despite liquid-liquid extraction and protein precipitation being examined as sample preparation methods, only protein precipitation demonstrated the acceptable results on almost 100% recoveries and no matrix effect. Moreover, the results on recoveries and matrix effect allowed us to use deionized water instead of analytes-free blood serum for the validation experiments. Thus, in this study we developed and validated a sensitive UPLC-MS/MS method for the simultaneous quantification of phenyl- and indole-containing metabolites. The main goal was to test the developed protocol for the analysis of the healthy volunteers’ serum and we suppose that this goal was successfully achieved as we managed to quantify most analytes.
The limitation of our method was mentioned in lines 494-495: “our subsequent research will attempt to reduce the LOQ levels of analytes in serum, which were not measured in this study (p-HBA, p-HPhPA, and 3ICA)” and in lines 470-475: “However, although the sample volume for this method is only 100 μL, there are studies that show the prospect of using even less matrix for analysis [22]. While it is not a disadvantage of the developed method for determination of analytes in human serum or plasma of adult people, it may be crucial in cases of children or neonates when the available volume of the sample might be limited.”
- Please provide some examples or references of how your method can be used in clinical practice or research. What are the potential applications and implications of your method?
The answer: We have mentioned this information in the Conclusion section “All data on the validation and analysis of the real samples is available in Supplementary Materials and can be used by the researchers in their future clinical studies as reference values. We believe that our protocol will be applied in clinics for the analysis of the diagnostically significant p-HPhLA and other potentially important aromatic metabolites of microbial and endogenous origin.” lines 553-558
Reviewer 4 Report
Comments and Suggestions for Authors
The manuscript submitted by Sobolev et al is devoted to development of a validated method for quantification of a series of phenyllactic and indoleacetic acids in human serum using HPLC-MS/MS. In my opinion, the study is very well designed, performed and represented as the manuscript. Having considered the submission, I have following comments:
1) Please mention here that it was 3IAA-d4 that served as the internal standard (either in the section 2.1 or 2.3).
2) Page 3, line 117: Did the authors mean HPLC-MS grade for acetonitrile?
3) Page 4, line 147: Did the authors mean volume-to-volume (v/v) ratio for formic acid instead of w/w?
4) Section 2.4.2, the title: The described method implies using an organic solvent for precipitation of proteins, therefore, the correct name of the method is protein precipitation.
5) Page 4, line 182: Why was this voltage used in the positive mode? It is possible to set 5500 V for IS in Sciex mass spectrometers, and the difference of 1500 V can be significant for sensitivity of the method.
Author Response
Reviewer 4
The manuscript submitted by Sobolev et al is devoted to development of a validated method for quantification of a series of phenyllactic and indoleacetic acids in human serum using HPLC-MS/MS. In my opinion, the study is very well designed, performed and represented as the manuscript. Having considered the submission, I have following comments:
1) Please mention here that it was 3IAA-d4 that served as the internal standard (either in the section 2.1 or 2.3).
The answer: We thank the reviewer for his high assessment of our work, for his time and suggested changes.We have added this information in the section 2.1 and 2.3 (lines 116, 140)
2) Page 3, line 117: Did the authors mean HPLC-MS grade for acetonitrile?
The answer: We have checked the information on the reagents and there is no mistake in their names: acetonitrile, HPLC-S gradient grade https://shop.biosolve-chemicals.eu/detail.php?id=1351
acetonitrile, LC-MS grade https://shop.biosolve-chemicals.eu/detail.php?id=1368
3) Page 4, line 147: Did the authors mean volume-to-volume (v/v) ratio for formic acid instead of w/w?
The answer: We thank the reviewer, it was a mistake and we corrected it (lines 155, 164).
4) Section 2.4.2, the title: The described method implies using an organic solvent for precipitation of proteins, therefore, the correct name of the method is protein precipitation.
The answer: We thank the reviewer, it should be “Protein precipitation” and we corrected it (line 166).
5) Page 4, line 182: Why was this voltage used in the positive mode? It is possible to set 5500 V for IS in Sciex mass spectrometers, and the difference of 1500 V can be significant for sensitivity of the method.
The answer: The influence of all ionization source parameters was studied using flow injection analysis (FIA) implemented on Sciex mass spectrometers. We mentioned it in Section 3.2 (lines 366-369). No significant difference in sensitivity was observed for the indole-containing analytes between 4000 V and 5500 V. For this reason, we set the ion spray voltage 4000 V.
Reviewer 5 Report
Comments and Suggestions for Authors
This study described the analysis of 4-Hydroxyphenyllactic Acid and some other diagnostically important metabolites of α-Amino acids in blood serum using HPLC-MS/MS method. The developed assay was validated and applied in quantification in healthy volunteers’ samples to reveal the reference range. Further authors claim that this method is comparable to previously reported GC-MS method. Overall, the manuscript is interesting and written well. Following comments to be addressed properly for further improvement of the manuscript.
1. Since the UPLC is used for separation, the term HPLC-MS/MS should be replaced by UPLC-MS/MS in title and throughout the manuscript.
2. Authors did not provide any demographical details and inclusion/exclusion criteria for the healthy volunteers participated in this study.
3. In sample preparation sample: both liquid liquid extraction and precipitation method was described, however authors claimed that protein precipitation as the sample preparation in abstract section. Kindly justify about it.
4. Author needs to provide the blank MRM chromatogram of target analytes including some chromatograms of actual healthy volunteers’ samples rather than ULOQ chromatogram in solvent in Figure 1.
5. Authors need to provide spectral representation of precursor to product ions of target compounds in supplementary data.
6. It is suggested to re-organize the conclusion section much better. The conclusion is lacking some basic components. It should be re-written in a well-structured manner. It should cover a summary of the problem(s), objectives methodology, findings, and recommendation(s).
Comments on the Quality of English Language
Minor correction required
Author Response
Reviewer 5
This study described the analysis of 4-Hydroxyphenyllactic Acid and some other diagnostically important metabolites of α-Amino acids in blood serum using HPLC-MS/MS method. The developed assay was validated and applied in quantification in healthy volunteers’ samples to reveal the reference range. Further authors claim that this method is comparable to previously reported GC-MS method. Overall, the manuscript is interesting and written well. Following comments to be addressed properly for further improvement of the manuscript.
- Since the UPLC is used for separation, the term HPLC-MS/MS should be replaced by UPLC-MS/MS in title and throughout the manuscript.
The answer: We thank the reviewer for his high assessment of our work, for his time and suggested changes. Also we thank the reviewer for this comment, it should be “UPLC-MS/MS” as it was mentioned in Section 2.5 “The Waters Acquity UPLC I-Class System” in line 173 and we corrected this throughout the text.
- Authors did not provide any demographical details and inclusion/exclusion criteria for the healthy volunteers participated in this study.
The answer: We have added this information in Section 2.2: “They were 19 women and 35 men aged from 20 to 67 years. At the time of the examination, volunteers were excluded from chronic liver and kidney diseases, and there were no general clinical signs of acute inflammation.” lines 130-132. Also this data was included in Supplementary Table S18.
- In sample preparation sample: both liquid liquid extraction and precipitation method was described, however authors claimed that protein precipitation as the sample preparation in abstract section. Kindly justify about it.
The answer: We have added this information in Abstract “The liquid-liquid extraction was also examined, but it demonstrated unsatisfactory results on recoveries and matrix effect.” (lines 22-24)
- Author needs to provide the blank MRM chromatogram of target analytes including some chromatograms of actual healthy volunteers’ samples rather than ULOQ chromatogram in solvent in Figure 1.
The answer: The chromatograms of the blank matrix and serum samples of healthy volunteers are presented in supplementary (Table S5) since placing them all in the main body of the article will take up a lot of space and make it harder to comprehend the text.
- Authors need to provide spectral representation of precursor to product ions of target compounds in supplementary data.
The answer: We have added this information in supplementary (Table S2) and mentioned it in Section 3.2 (lines 365-366)
- It is suggested to re-organize the conclusion section much better. The conclusion is lacking some basic components. It should be re-written in a well-structured manner. It should cover a summary of the problem(s), objectives methodology, findings, and recommendation(s).
The answer: We have changed the Conclusion section according to the suggested comments: “Phenyl- and indole-containing metabolites of tyrosine, phenylalanine and tryptophan are potential biomarkers of different disorders and their simultaneous determination in human blood serum is a challenging issue because of their different chemical structures and endogenous content in human blood. All analytes were separated on a C18 chromatography column except two isobaric compounds p-HPhPA and PhLA. However, they had different product ions and were successfully analyzed using MRM-mode. Despite liquid-liquid extraction and protein precipitation being examined as sample preparation methods, only protein precipitation demonstrated the acceptable results on almost 100% recoveries and no matrix effect. Moreover, the results on recoveries and matrix effect allowed us to use deionized water instead of analytes-free blood serum for the validation experiments. Thus, in this study we developed and validated a sensitive UPLC-MS/MS method for the simultaneous quantification of phenyl- and indole-containing metabolites. The main goal was to test the developed protocol for the analysis of the healthy volunteers’ serum and we suppose that this goal was successfully achieved as we managed to quantify most analytes. All data on the validation and analysis of the real samples is available in Supplementary Materials and can be used by the researchers in their future clinical studies as reference values. We believe that our protocol will be applied in clinics for the analysis of the diagnostically significant p-HPhLA and other potentially important aromatic metabolites of microbial and endogenous origin.